# Aggregation-Morphology-Dependent Electrochemical Performance of Co_3_O_4_ Anode Materials for Lithium-Ion Batteries

**DOI:** 10.3390/molecules24173149

**Published:** 2019-08-29

**Authors:** Linglong Kong, Lu Wang, Deye Sun, Su Meng, Dandan Xu, Zaixin He, Xiaoying Dong, Yongfeng Li, Yongcheng Jin

**Affiliations:** 1State Forestry and Grassland Administration Key Laboratory of Silviculture in downstream areas of the Yellow River, Shandong Agricultural University, No. 61 Daizong Road, Taian 271018, China; 2Qingdao Institute of Bioenergy and Bioprocess Technology, Chinese Academy of Sciences, No.189 Songling Road, Qingdao 266101, China; 3School of Materials Science and Engineering, Nankai University, Tianjin 300350, China

**Keywords:** lithium-ion batteries, anode materials, cobalt oxides, aggregation morphology, electrochemical performance

## Abstract

The aggregation morphology of anode materials plays a vital role in achieving high performance lithium-ion batteries. Herein, Co_3_O_4_ anode materials with different aggregation morphologies were successfully prepared by modulating the morphology of precursors with different cobalt sources by the mild coprecipitation method. The fabricated Co_3_O_4_ can be flower-like, spherical, irregular, and urchin-like. Detailed investigation on the electrochemical performance demonstrated that flower-like Co_3_O_4_ consisting of nanorods exhibited superior performance. The reversible capacity maintained 910.7 mAh·g^−1^ at 500 mA·g^−1^ and 717 mAh·g^−1^ at 1000 mA·g^−1^ after 500 cycles. The cyclic stability was greatly enhanced, with a capacity retention rate of 92.7% at 500 mA·g^−1^ and 78.27% at 1000 mA·g^−1^ after 500 cycles. Electrochemical performance in long-term storage and high temperature conditions was still excellent. The unique aggregation morphology of flower-like Co_3_O_4_ yielded a reduction of charge-transfer resistance and stabilization of electrode structure compared with other aggregation morphologies.

## 1. Introduction

Recently, secondary batteries, especially lithium-ion batteries (LIBs) have become significant components in portable devices, electric vehicles, and exploitation systems of clean energy, due to their satisfactory energy density, lifetime, and environmental friendliness [1,2]. However, the booming society is imposing a strong demand on energy density of LIBs [3,4]. The current LIBs utilizing traditional electrode materials, such as lithium cobalt oxides and graphite, can only achieve limited enhancement of energy density in the range of 150–300 Wh·kg^−1^ [5]. Thus, exploiting high-capacity electrode materials has been the primary solution [6]. Regarding anodes, transition metal oxides, such as NiO [7], Fe_2_O_3_ [8], ZnO [9], CuO [10], MnO [11], Co_3_O_4_ [12], ZnCo_2_O_4_ [13], and NiCo_2_O_4_ [14], are promising future anode materials to substitute for the widely used carbon materials. Typically, Co_3_O_4_, with its high theoretical capacity (890 mAh·g^−1^) owing to storage for eight lithium ions per molecule based on the reversible conversion reaction between Co_3_O_4_/Li and Co/Li_2_O (Co_3_O_4_ + 8Li ↔ 3Co + 4Li_2_O), has been an important candidate as a next generation anode. Additionally, high volumetric capacity of Co_3_O_4_ over graphite is another crucial advantage [15].However, the intrinsic low electrical conductivity, poor Coulombic efficiency [16], high voltage hysteresis [17], high average delithiation potential [15], and high large volume fluctuation in the discharge/charge procedure induce severe pulverization and electrochemical instability, and thus impair the capacity stability and power performance.

Numerous effective solutions have been developed to solve the existing issues of Co_3_O_4_ anode materials. They can be classified in three types: (i) rational design of the microstructure, (ii) surface or bulk modification, and (iii) constructing composites. The microstructure of Co_3_O_4_ can be designed as nanorods [18], nanotubes [19], nanocubes [20], nanosheets [21], nanowires [22], microspheres [23], polyhedra [24], fusiform [25], and 3D structures, such as dumbbells [26], hierarchical networks [27], hierarchical arrays [28], mesoporous octahedra [29], and multishelled spheres [30]. Nano- and microscale Co_3_O_4_ structures can alleviate the volume expansion/shrink, enhance the contact of particles and electrolyte, and shorten the pathway of ions and electrons. Modification of Co_3_O_4_ by coating with carbon [31,32], conductive polymer [33], and MoS_2_ [34], as well as by doping C [35], K [36], Ni, and Zn [37], has been demonstrated to inhibit the structure and capacity decay. Similarly, fabricating Co_3_O_4_ composites with CNTs [38], graphene [39], metal oxides [40,41], and porous carbon [42] through in situ or ex situ methods has contributed to the improvement of its electrochemical stability. It has been positively confirmed that the structural characteristics, such as particle size, elemental distribution, and aggregation morphology, largely influence the electrochemical performance of Co_3_O_4_. Nano-scale Co_3_O_4_ particles can effectively solve the problem of conductivity and volume effect, but their high surface area may also cause undesirable aggregation, large interparticle resistance, and excessive loss of lithium in forming the solid electrolyte interphase (SEI) [24]. Bottom-up assembly based on nanoparticles to microscale aggregations may avoid these faults, and make Co_3_O_4_ more available as a commercial micrometer electrode material. Although many works have synthesized diverse Co_3_O_4_ structures with nano primary particles, the main focus has mostly centered on improved electrochemical properties, and the detailed study on the structure–function relationship between electrochemical performance and aggregation morphology to date is not sufficient.

Herein, four kinds of precursors with different morphology were prepared by mild coprecipitation reaction using different cobalt sources. Co_3_O_4_ anode materials in distinct aggregations (flower-like, spherical, irregular, and urchin-like) were obtained after calcination. The systematic testing results revealed that flower-like Co_3_O_4_ materials possessed superior electrochemical stability, especially at higher current densities. The reversible capacity maintained 910.7 mA·g^−1^ at 500 mA·g^−1^ and 717 mAh·g^−1^ at 1000 mA·g^−1^ after the 500th cycle, respectively. The superior electrochemical behavior of flower-like Co_3_O_4_ derived from its structural stability and lower reaction impedance.

## 2. Experimental Results and Discussion

The aggregation morphology of Co_3_O_4_ materials was flexibly regulated by constructing various precursors, as shown in Figure 1. Specifically, the homogeneous precipitation method was applied in mild conditions, using urea as precipitant and cobalt salts as cobalt sources. SEM images of the synthesized precursors proved the successful regulation of the morphology (Figure 2a). It was proven that different cobalt salts produced precursors in various shapes. Bunches of flower-like precursors consisting of nanorods were inclined to generate when using cobalt chlorate. The configuration resulting from a cobalt sulfate precursor was spherical. Micro-clusters made of flakes tended to form for cobalt acetate. Urchin-like particles were prepared by applying cobalt nitrate. The formation process of the precursors involves nuclei and growth, which can be largely influenced by the microenvironment of the reaction solution. The CO_3_^2−^ and OH^−^ produced from the decomposition of urea contributed to the initial nuclei, and the formed crystal nuclei were inclined to adsorb the existing anions in the solution, leading to a negatively charged surface [43]. The existing Cl^−^, SO_4_^2−^, AC^−^, and NO_3_^−^ may then exert influence on the following growth behavior based on electrostatic interactions [43]. Thus, the morphology of the precursors corresponding to different cobalt sources could be easily adjusted. The crystal structures of the obtained precursors originating from different cobalt salts were confirmed and are shown in Figure 2b. The XRD pattern of the precursor from cobalt chlorate demonstrated fine crystallinity, a good match with the Co(CO_3_)_0.35_Cl_0.20_(OH)_1.10_ (JCPDS card No. 38–0547), and no apparent impurity. The precursor prepared by cobalt sulfate possessed similar crystal structure, but the crystallinity seemed relatively low. As to the precursors corresponding to cobalt acetate and cobalt nitrate, the characteristic diffraction peaks were well indexed to Co(CO_3_)_0.5_(OH) 0.11H_2_O (JCPDS card No. 48–0083). The obvious peaks at 2θ = 14.44° and 24.03° may be assigned to CoC_2_O_4_ (JCPDS card No. 37–0719). Thermal behavior analysis of the as-prepared precursors was conducted by TG and is displayed in Figure 2c. During the testing process, the main weight loss occurred in the range of 200–300 °C, and the residual weight remained stable until 550 °C. The total loss in weight was calculated to be 26.2%, 26.3%, 25.3%, and 27.7%, respectively, values which are basically consistent with the theoretical values [40,41]. Thus, the calcination temperature was fixed at 550 °C, and the precursors transformed into oxides.

After calcinating at 550 °C for 2 h, the acquired samples had the structure shown in Figure 3a. All the diffraction peaks corresponded to the characteristic peaks of cubic Co_3_O_4_ (JCPDS card No. 09–0418), revealing the generation of Co_3_O_4_ with the expected crystallinity and no impurity. The lattice constants (a) of Co_3_O_4_-Cl, Co_3_O_4_-SO_4_, Co_3_O_4_-AC, and Co_3_O_4_-NO_3_ were 8.087 Å, 8.076 Å, 8.070 Å, and 8.060 Å, respectively, which were close to the value (a = 8.084 Å) of the standard Co_3_O_4_ (Figure 3b). Moreover, the average particle size of the as-prepared Co_3_O_4_ could be calculated based on XRD data and Scherrer equation. Co_3_O_4_-Cl possessed a larger particle size of 27.8 nm than that of Co_3_O_4_-SO_4_ (16.9 nm), while the values for Co_3_O_4_-AC and Co_3_O_4_-NO_3_ were identical (25.2 nm). FT-IR and Raman spectra further verified the formation of the desired oxides. The absorption peaks at 576 and 662 cm^−1^ in Figure 3c stand for the characteristic stretching vibration peaks of Co–O band, and the typical peaks seen at 464, 507, 604, and 674 cm^−1^ (Figure 3d) belong to crystalline Co_3_O_4_ [44,45], which provided sufficient evidence for successful synthesis.

Aggregation morphology of the Co_3_O_4_ obtained with different cobalt salts was investigated by SEM. The images in Figure 4 present the distinct differences. The morphology of the calcinated products was maintained to some degree from the precursors, but obvious changes occurred. For Co_3_O_4_-Cl, flower-like particles were assembled into bamboo-like rods, and the rods consisted of nanograins (Figure 4a,b). Spherical precursors of Co_3_O_4_-SO_4_ were sintered into ellipsoids (Figure 4c), which consisted of numerous small grains (Figure 4d). Flake-like precursors corresponding to Co_3_O_4_-AC were transformed to monodispersed nanoparticles, but serious aggregation occurred and led to irregular clusters (Figure 4e,f). The urchin-like shape still existed for Co_3_O_4_-NO_3_ (Figure 4g), but the nanorods changed into bamboo-like rods (Figure 4h). Using the EDS detecting system, the distributions of elemental oxygen and cobalt were acquired for the as-prepared Co_3_O_4_ with different micromorphologies, as shown in Figure 5. The results showed that elemental O and Co were distributed homogenously in the various aggregates. The above results demonstrate that Co_3_O_4_ materials with different aggregation morphologies can be successfully prepared by the rational use of cobalt sources in the synthesis process.

The relationship between the aggregation states of Co_3_O_4_ and its performance were illustrated in detail by electrochemical analysis. Cyclic voltammograms (CV) were first collected in the coin cells from 0.01 V to 3.0 V, and shown in Figure 6. In the initial negative scanning process, the obvious cathodic peaks located around 0.80 V were ascribed to the transformation from Co_3_O_4_ to Co and some side reactions, including the generation of the SEI layer [46,47] The reduction potentials for Co_3_O_4_-Cl, Co_3_O_4_-SO_4_, Co_3_O_4_-AC, and Co_3_O_4_-NO_3_ were 0.854 V, 0.765 V, 0.850 V, and 0.817 V (Table 1), respectively, and the difference in potential may indicate that small grains benefited the electrochemical reaction. In the subsequent positive scanning, broad anodic peaks near 2.1 V occurred, and were related to the transition of Co and Li_2_O [48]. The corresponding oxidation potentials were 2.055 V, 2.118 V, 2.061 V, and 2.070 V (Table 1), respectively. The slight increase of anodic potential for Co_3_O_4_-SO_4_ may have been caused by excessive formation of SEI on the smaller primary particles. As the CV test proceeded, the potential for reduction reaction inclined to shift to 1.1 V, and the values of Co_3_O_4_-SO_4_ displayed a larger increase, while the oxidation potential stayed at almost 2.1 V. The tested curves were basically overlapped, revealing preferable reversibility during the electrochemical conversion. The CV results demonstrate that Co_3_O_4_ materials exhibit similar electrochemical behavior in spite of particle morphology.

The practical capacity for lithium storage of the as-prepared Co_3_O_4_ was evaluated through galvanostatic test. The relevant results are listed in Figure 7. The initial discharge–charge curves at 200 mA·g^−1^, shown in Figure 7a, showed a similar rule to the cathodic/anodic potentials of the CV data. The plateau potential in discharging of to Co_3_O_4_-SO_4_ was slightly lower, combined with a larger capacity (1281.8 mAh·g^−1^). The initial coulombic efficiencies (CE) of Co_3_O_4_-Cl, Co_3_O_4_-SO_4_, Co_3_O_4_-AC, and Co_3_O_4_-NO_3_ were close, and irreversible capacity can occur due to undesired side reactions. It seemed that the spherical Co_3_O_4_-SO_4_ with smaller grains displayed superior electrochemical performance. Long-term cycling was conducted at 200 mA·g^−1^, 500 mA·g^−1^, and 1000 mA·g^−1^. Figure 7b reveals that Co_3_O_4_-SO_4_ delivered a higher capacity in the first 30 cycles, which demonstrates the advantage of decreasing grain size compared with morphology. The capacity then showed abrupt decay, which may have been due to the collapse of secondary particles. By contrast, Co_3_O_4_-Cl, Co_3_O_4_-AC, and Co_3_O_4_-NO_3_ possessed stable cyclic traits. The capacity of Co_3_O_4_-Cl was relatively smaller than those of Co_3_O_4_-AC and Co_3_O_4_-NO_3_ during the 100 cycles, which indicates that grain size may be the key at low current density, regardless of aggregation morphology. Reaction sites can be maximized along with the reduction on particle size. Obviously, the discharged capacities for Co_3_O_4_-Cl, Co_3_O_4_-AC, and Co_3_O_4_-NO_3_ samples were still higher than the theoretical value even after 100 cycles, and the extra capacity be in part due to surface pseudocapacitive behavior, which is commonly discovered in transition metal oxides with high surface area and nano size [47]. Moreover, the reversible capacities at 200 mA·g^−1^ for Co_3_O_4_-Cl, Co_3_O_4_-AC, and Co_3_O_4_-NO_3_ samples increased along with the cycling, originating from the progressive activation and gradual formation of a gel-like surface film [47,48,49,50]. Cycle properties at higher currents (500 mA·g^−1^ and 1000 mA·g^−1^) were then analyzed. At 500 mA·g^−1^ (Figure 7c), spherical Co_3_O_4_-SO_4_ materials still had superior capacity before 30 cycles, then the capacity sharply decreased and fluctuated. By contrast, the flower-like Co_3_O_4_-Cl sample showed preferable cycle stability. The capacity remained at 910.7 mAh·g^−1^ after 500 cycles, and the capacity retention rate reached to 92.7% with only tiny decay (0.015% per cycle). Co_3_O_4_-AC clusters and urchin-like Co_3_O_4_-NO_3_ performed unstably in the long-term cycle. The same situation appeared at the current density of 1000 mA·g^−1^ (Figure 7d). The Co_3_O_4_-Cl electrode achieved a capacity of 717 mAh·g^−1^ in the 500th cycle, with an acceptable retention rate (78.27%). Admittedly, nanoparticles benefitted from a reducing path length of Li^+^, but side reactions were also enhanced [43]. In addition, the volume changes may have resulted in the irreversible destruction of secondary particles. Therefore, the flower-like Co_3_O_4_-Cl with relatively larger grains displayed more satisfactory stability, especially at high current density.

The rate performance in Figure 8a and low-temperature test in Figure 8c also demonstrated that Co_3_O_4_ with small grains (Co_3_O_4_-SO_4_) had advantages on capacity, but the capacity stability seemed slight deficient. As to the cells with a standing time of 60 days (Figure 8b), the reversible capacity at 500 mA·g^−1^ decreased compared to the cells without standing (Figure 7c). Co_3_O_4_-Cl anode materials with flower-like morphology possessed superior cyclic stability. The capacity reached 695.9 mAh·g^−1^ after 300 cycles, indicating the structural advantages of Co_3_O_4_-Cl. Cells working under extreme conditions, such as high temperatures, can induce aggravating side reactions. Transition metal oxides anodes have the same problem. The test results at 45 °C and 500 mA·g^−1^ proved the enhanced cycle stability of flower-like Co_3_O_4_ anode materials, and the relevant capacity reached 947.6 mAh·g^−1^ in the 100th cycle (Figure 8d).

Electrochemical kinetics and reaction mechanism were then investigated by implementing CV tests at different scan rates from 0.1 mV·s^−1^ to 1.0 mV·s^−1^ (Figure 9a–d). The relationship between peak current (I_p_) and scan rate (ν) can be described with Equation (1) to determine the contributions of diffusion-limited and capacitive effects during the cathodic and anodic process.
I_p_ = aν^b^(1)
where a and b refer to variable parameters. B = 1 implies capacitive behavior, while b = 0.5 corresponds to a diffusion-controlled process. The specific value of b can be calculated by fitting the line of logI_p_ vs. logν and obtaining the slope. The detailed data for cathodic and anodic peaks are shown in Figure 9e,f, including the fitted slopes. The b values of the Co_3_O_4_-SO_4_ sample were 0.764 and 0.827, respectively, related to the cathodic and anodic process, which were higher than those of other Co_3_O_4_ samples. A higher b value reflects the enhanced capacitive effect and improved rate property [51]. In the same way, the lithium ion diffusion property can also be calculated according to Equation (2).
I_p_ = (2.69 × 10^5^)n^1.5^aD^0.5^ν^0.5^ΔC_0_(2)
where I_p_ represents peak current, n relates to the number of electrons per reactant, a corresponds to the area of active electrode, D denotes lithium ion diffusion coefficient, and ΔC_0_ refers to the concentration change of Li during electrochemical reaction. Considering the constant parameters of n, a, and ΔC_0_, the slope for the line between I_p_ and square root of the scan rate can illustrate the lithium ion diffusion property. The relevant plots are displayed in Figure 9g,h. The results by linear fitting demonstrated that spherical Co_3_O_4_ with small grains (Co_3_O_4_-SO_4_) had higher D values in both cathodic and anodic processes. In comparison, flower-like Co_3_O_4_ with nanorods (Co_3_O_4_-Cl) possessed inferior lithium ion diffusion.

Since the spherical Co_3_O_4_ with small grains (Co_3_O_4_-SO_4_) had stronger Li^+^ diffusion ability than that of flower-like Co_3_O_4_ with nanorods (Co_3_O_4_-Cl), the relevant electrochemical performance should be superior. However, the above electrochemical results reveal that flower-like Co_3_O_4_ exhibited more excellent performance. Therefore, it can be concluded that grain size plays an important role in delivering capacity, and aggregation morphology decides the cycle performance under various working conditions. Irregular clusters with nanoparticles (Co_3_O_4_-AC) may promote excessive side reactions. Spherical Co_3_O_4_ with small grains (Co_3_O_4_-SO_4_) may easily suffer from the fracture of secondary particles, as might urchin-like Co_3_O_4_ with nanorods (Co_3_O_4_-NO_3_). Flower-like Co_3_O_4_ with nanorods (Co_3_O_4_-Cl) may preferably balance stability of structure and capacity.

Further measurement on the cycled electrodes by EIS and SEM provided more proof of the performance difference. EIS spectra (Figure 10) were collected on fresh electrodes and cycled electrodes after 500 cycles at 500 mA·g^−1^ and 1000 mA·g^−1^. All the spectra comprised a semicircle (medium–high frequency) and an inclined line (low frequency region) before and after cycling. The charge-transfer resistance corresponding to the semicircle of the cycled Co_3_O_4_-Cl electrodes was obviously smaller than those of other electrodes at 500 mA·g^−1^ and 1000 mA·g^−1^. The reduced resistance to charge transfer facilitates rapid electrochemical reaction. Meanwhile, the active particles were clearly seen on the fresh electrodes (Figure 11a–d). After 500 cycles at 500 mA·g^−1^, the surface on the electrode was covered with SEI films, which has also been detected in other studies [47]. The surface for the Co_3_O_4_-Cl electrode remained intact (Figure 11e), while pores and fractures emerged on the electrodes of Co_3_O_4_-SO_4_, Co_3_O_4_-AC, and Co_3_O_4_-NO_3_. In turn, this indicates the enhanced structural stability of the electrode, which may be radically ascribed to the active materials. Flower-like Co_3_O_4_ with nanorods (Co_3_O_4_-Cl) exhibited advantages in constructing a stable electrode with superior performance. Therefore, aggregation morphology of the Co_3_O_4_ materials played a vital role in achieving high performance and stability.

By utilizing the prepared Co_3_O_4_ as anode active materials, full batteries were assembled to evaluate a realistic application scenario. The cathodes were composed of commercial NCM811 with an areal loading of 4.0–5.0 mg·cm^−1^. The working voltage window was set at 0.01–4.3 V, and the specific capacity was calculated according to the weight of NCM. Moreover, the capacity of the Co_3_O_4_ electrode was almost equal to that of the positive electrode, and moderate pre-lithiation on the Co_3_O_4_ negative electrode was adopted before assembling full battery. Figure 12a reveals the initial charge–discharge curves of the full batteries at 40 mA·g^−1^. The inclined charging and discharging platforms in the full batteries were similar to those of the NCM/Li half-cell. Figure 12b presents the cycle performance of the assembled full batteries at 100 mA·g^−1^. The cycle stability was relatively good, and the capacities of the batteries using Co_3_O_4_-Cl, Co_3_O_4_-SO_4_, Co_3_O_4_-AC, and Co_3_O_4_-NO_3_ remained at 137.2, 124.3, 126.4, and 107.1 mAh·g^−1^, respectively, after 20 cycles. The full battery adopting the flower-like Co_3_O_4_ delivered a comparatively higher capacity, indicating the structural advantage of flower-like Co_3_O_4_ in practical applications.

## 3. Experimental Materials and Methods

### 3.1. Experimental Materials

The chemical reagents used, including cobalt chlorate hexahydrate (AR, ≥ 99.0%), cobalt sulfate heptahydrate (AR, ≥ 99.5%), cobalt acetate tetrahydrate (99.9% metal basis), cobalt nitrate hexahydrate (AR, 99.0%), and urea (AR, 99.0%) were purchased from Sinopharm Chemical Reagent Co., Ltd. (Shanghai, China) and used without further purification.

### 3.2. Experimental Methods

#### 3.2.1. Synthesis of the Precursors

The precursors were prepared using the homogeneous precipitation method [52]. In the typical preparation process, 0.004 mol cobalt salts and 0.02 mol urea were weighed and transferred into a 100 mL Teflon liner with 50 mL distilled water, followed by 15 min ultrasonic treatment. The liner was then placed in a stainless steel autoclave and heated at 95 °C for 8 h. The precipitates in the cooling autoclave were separated, washed repeatedly with distilled water and ethanol, and dehydrated at 60 °C for 12 h. Four kinds of precursors were used, corresponding to four cobalt salts, respectively, and the preparation procedure was the same.

#### 3.2.2. Synthesis of Cobalt Oxides

The dried precursors were calcined in the muffle furnace under air atmosphere. The sintering temperature and holding time were set at 550 °C and 2 h, respectively. The heating rate was fixed at 5 °C·min^−1^. The black cobalt oxide powders were obtained after the furnace cooling to ambient temperature. The four cobalt oxides were marked as Co_3_O_4_-Cl, Co_3_O_4_-SO_4_, Co_3_O_4_-AC, and Co_3_O_4_-NO_3_, respectively.

#### 3.2.3. Material Characterization

The crystal structure, morphologies, and elemental distribution of the synthesized materials and electrodes were characterized by X-ray diffraction (XRD, Bruker AXS D8, Karlsruhe, Germany) at 4°·min^−1^, and scanning electron microscopy (SEM, Hitachi S-4800, Tokyo, Japan) combined with EDS detector. The chemical states of the obtained materials were tested by applying Fourier transformation infrared spectroscopy (FT-IR, Bruker V80, Karlsruhe, Germany) at 4000–400 cm^−1^ and Raman spectroscopy (Thermo Scientific DXR, Waltham, MA, USA, λ = 532 nm). Thermal behavior analysis was carried out in air using a thermal gravimetric analyzer (TGA, Thermo Scientific, Waltham, MA, USA) at 10 °C·min^−1^ from 30 °C to 800 °C.

#### 3.2.4. Electrochemical Measurements

Electrochemical measurements of the prepared anode materials were carried out in 2032-type coin cells, including working electrodes, separators (Celgard 2400, Φ = 17 mm), counter electrodes (Li foils, Φ = 16 mm), and electrolyte (1 M LiPF6 in EC/DEC, EC: DEC = 3:7, mass ratio). The working electrodes (Φ = 12 mm) were fabricated by blending active materials, Super P and poly(vinylidene fluoride) (70:20:10, weight ratio) in the dispersant (*N*-methyl-2-pyrrolidone, NMP) for 4 h, coating the slurry on the copper foil, evaporating the solvent in the oven, and subsequently punching into disks. The mass loading of active materials was controlled at 0.7–1.0 mg·cm^−1^. The cells were assembled in the glove-box and aged for 4 h before testing. Galvanostatic discharge/charge results were acquired using Land instruments (CT-2001A) within the given potential region (0.01–3.0 V, vs. Li/Li^+^). Cyclic voltammograms (CV) and electrochemical impedance spectra (EIS) were conducted on an electrochemical workstation (Bio-logic VSP) with a set potential window (0.01–3.0 V, vs. Li/Li^+^) and frequency range (10 mHz–100 kHz).

## 4. Conclusions

In summary, the aggregation morphology of the Co_3_O_4_ materials can be easily regulated by synthesizing precursors in various shapes with different cobalt sources in the coprecipitation process, including spherical, flower-like, irregular, and urchin-like Co_3_O_4_. Applied in a battery system, flower-like Co_3_O_4_ with nanorods displayed a superior reversible capacity of 910.7 mAh·g^−1^ at 500 mA·g^−1^ and 717 mAh·g^−1^ at 1000 mA·g^−1^ after 500 cycles, enhanced cyclic stability with a capacity retention rate of 92.7% at 500 mA·g^−1^ and 78.27% at 1000 mA·g^−1^ after 500 cycles, and high performance in harsh environments (long-term storage and high temperature). Although the capacitive character and diffusion property of flower-like Co_3_O_4_ were slightly poor, the lower charge-transfer resistance and stable electrode structure owing to the advantage of its unique aggregation morphology contributed a satisfying performance. When constructing Co_3_O_4_-based anode materials, the property of single grains and aggregates should be simultaneously considered, especially the morphology of the aggregates. Thus, the relationship between aggregation morphology and performance of Co_3_O_4_ can be applied to synthesize other high-performance anode materials for LIBs.

## Figures and Tables

**Figure 1 molecules-24-03149-f001:**
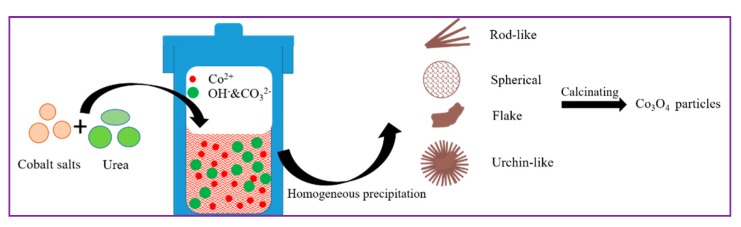
Schematic illustrations of synthesizing precursors and calcinated products.

**Figure 2 molecules-24-03149-f002:**
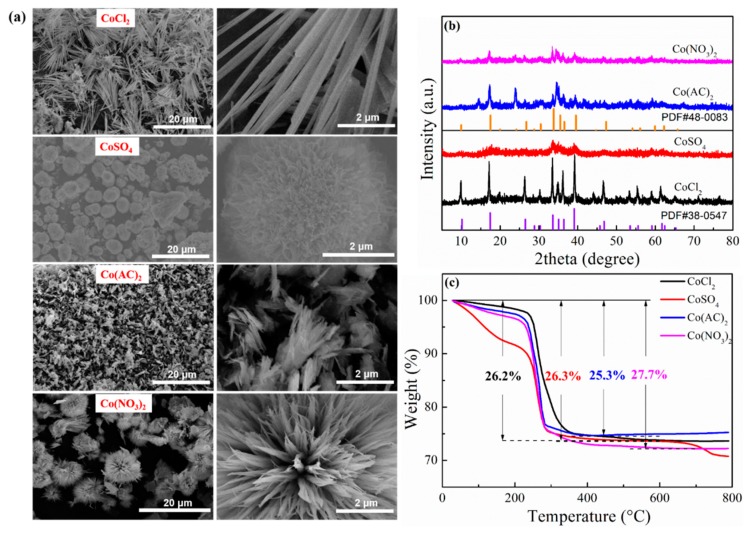
(**a**) Scanning electron microscope (SEM) images of the precursors obtained with cobalt chloride, cobalt sulfate, cobalt acetate, and cobalt nitrate as cobalt sources. (**b**) X-ray diffraction (XRD) patterns of the prepared precursors and (**c**) thermogravimetric (TG) curves of the synthesized precursors with different cobalt sources.

**Figure 3 molecules-24-03149-f003:**
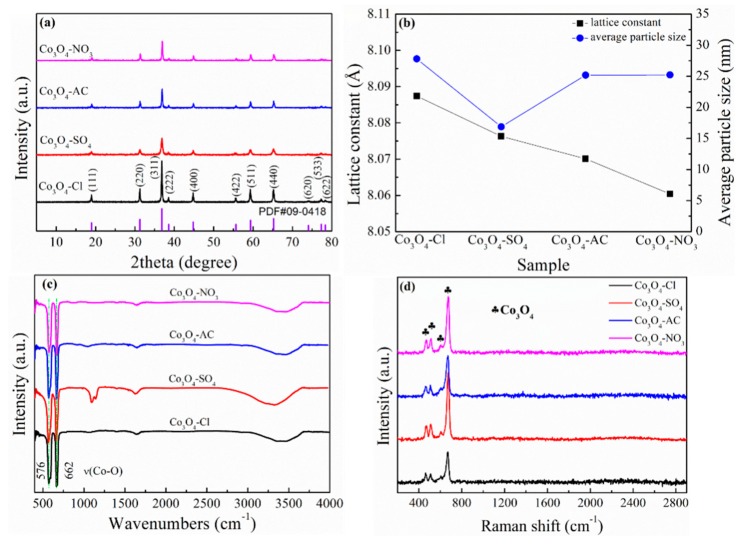
(**a**) XRD patterns of the calcinated products combined with the (**b**) lattice constant/average particle size, (**c**) Fourier transform infrared (FT-IR) spectra, and (**d**) Raman spectra.

**Figure 4 molecules-24-03149-f004:**
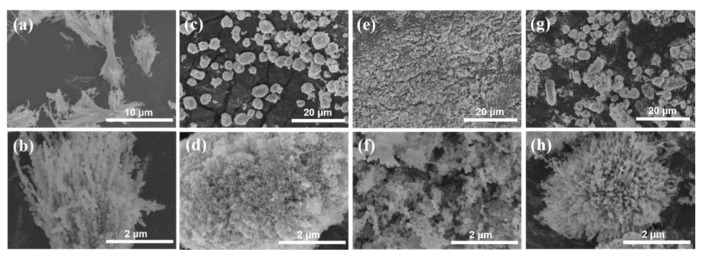
SEM images of the calcinated products related to (**a**,**b**) Co_3_O_4_-Cl, (**c**,**d**) Co_3_O_4_-SO_4_, (**e**,**f**) Co_3_O_4_-AC, and (**g**,**h**) Co_3_O_4_-NO_3_.

**Figure 5 molecules-24-03149-f005:**
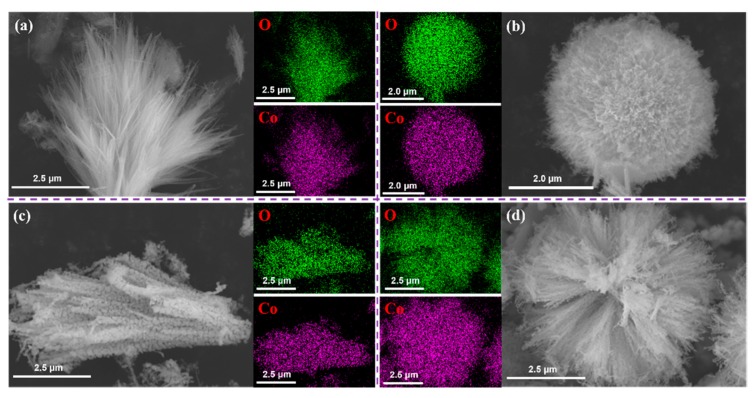
Distribution of elemental cobalt and oxygen corresponding to (**a**) Co_3_O_4_-Cl, (**b**) Co_3_O_4_-SO_4_, (**c**) Co_3_O_4_-AC, and (**d**) Co_3_O_4_-NO_3_.

**Figure 6 molecules-24-03149-f006:**
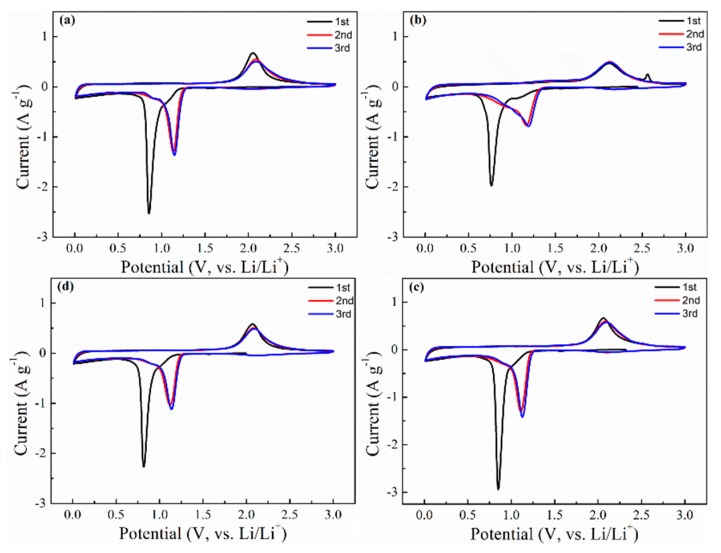
CV curves of (**a**) Co_3_O_4_-Cl, (**b**) Co_3_O_4_-SO_4_, (**c**) Co_3_O_4_-AC, and (**d**) Co_3_O_4_-NO_3_.

**Figure 7 molecules-24-03149-f007:**
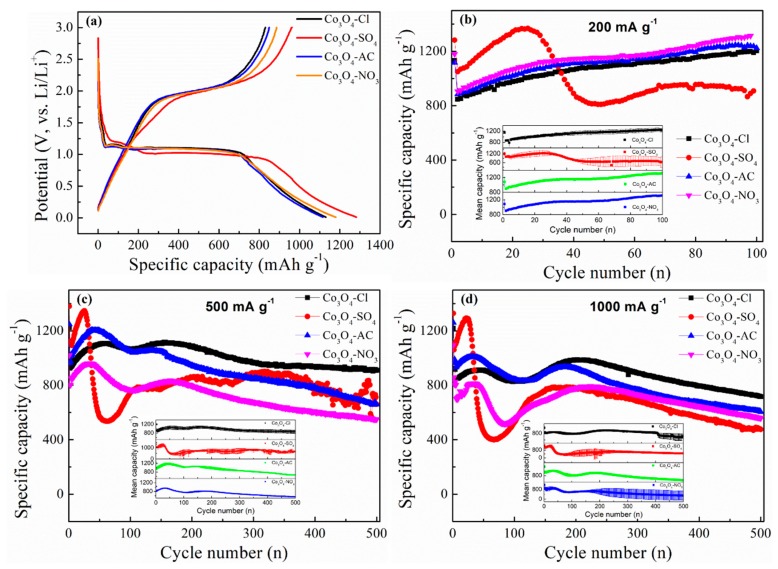
(**a**) Initial discharge–charge curves of Co_3_O_4_ at 200 mA·g^−1^. Cycle performance of different Co_3_O_4_ at the current density of (**b**) 200 mA·g^−1^, (**c**) 500 mA·g^−1^, and (**d**) 1000 mA·g^−1^. The insert pictures represent the error bar with standard deviation. The specific capacity corresponds to the discharge process, i.e., Co_3_O_4_ reacts with Li^+^ to form Co and Li_2_O.

**Figure 8 molecules-24-03149-f008:**
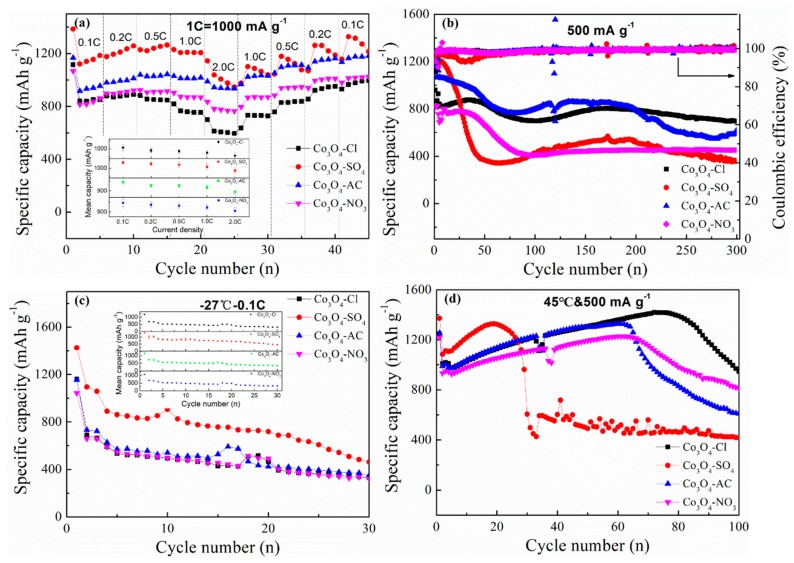
(**a**) Rate performances of the as-prepared Co_3_O_4_-Cl, Co_3_O_4_-SO_4_, Co_3_O_4_-AC, and Co_3_O_4_-NO_3_ at various current densities from 0.1 C to 2.0 C. (**b**) Cycle stability of Co_3_O_4_-Cl, Co_3_O_4_-SO_4_, Co_3_O_4_-AC, and Co_3_O_4_-NO_3_ materials in the cells standing for 60 days and cycled at 500 mA·g^−1^. (**c**,**d**) Cycle performance of the cells at low temperature (−27 °C) and high temperature (45 °C). The insert pictures represent the error bar with standard deviation. The specific capacity corresponds to the discharge process, i.e., Co_3_O_4_ reacts with Li^+^ to form Co and Li_2_O.

**Figure 9 molecules-24-03149-f009:**
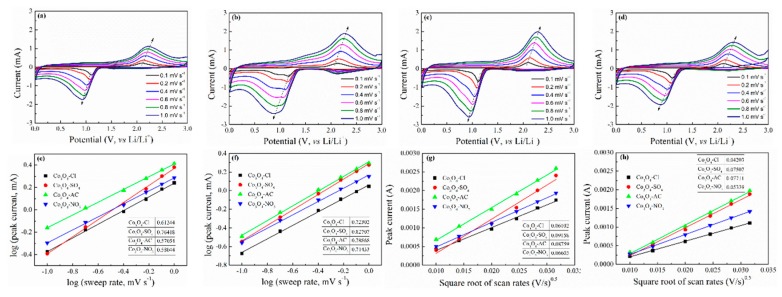
CV curves of (**a**) Co_3_O_4_-Cl, (**b**) Co_3_O_4_-SO_4_, (**c**) Co_3_O_4_-AC, and (**d**) Co_3_O_4_-NO_3_ at different scan rates (0.1 mV·s^−1^, 0.2 mV·s^−1^, 0.4 mV·s^−1^, 0.6 mV·s^−1^, 0.8 mV·s^−1^, and 1.0 mV·s^−1^). The relationship between peak current and scan rate in the (**e**) cathodic and (**f**) anodic process determines the value b in the equation (I = aν^b^). Plot of peak current of (**g**) cathodic reaction and (**h**) anodic reaction versus the square root of scan rates.

**Figure 10 molecules-24-03149-f010:**
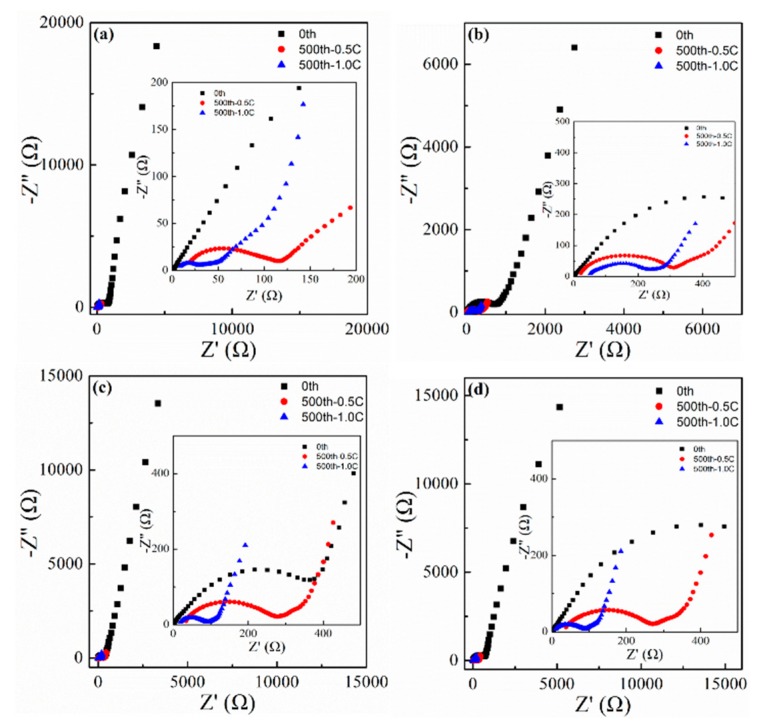
Electrochemical impedance spectra (EIS) spectra for the cells employing (**a**) Co_3_O_4_-Cl, (**b**) Co_3_O_4_-SO_4_, (**c**) Co_3_O_4_-AC, and (**d**) Co_3_O_4_-NO_3_ before and after cycling.

**Figure 11 molecules-24-03149-f011:**
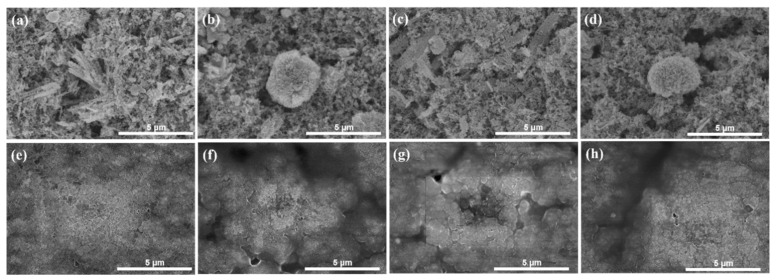
SEM images of the (**a**–**d**) fresh electrode and (**e**–**h**) cycled electrode (500th cycle at 500 mA·g^−1^) using (**a**,**e**) Co_3_O_4_-Cl, (**b**,**f**) Co_3_O_4_-SO_4_, (**c**,**g**) Co_3_O_4_-AC, and (**d**,**h**) Co_3_O_4_-NO_3_ as active materials.

**Figure 12 molecules-24-03149-f012:**
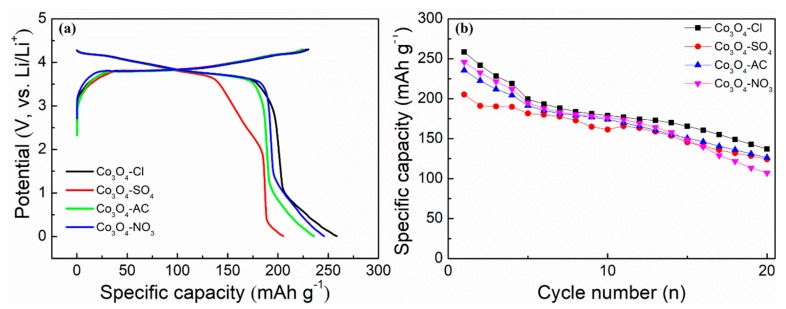
(**a**) The initial charge–discharge profiles of the full batteries at 40 mA·g^−1^, and (**b**) cycle performance of the full batteries using NCM811 cathode materials and the synthesized Co_3_O_4_ anode materials at 100 mA·g^−1^.

**Table 1 molecules-24-03149-t001:** The data originating from the CV curves in Figure 6.

Sample	1st Cycle		2nd Cycle		3rd Cycle	
	E_R_ (V)	E_O_ (V)	E_R_ (V)	E_O_ (V)	E_R_ (V)	E_O_ (V)
Co_3_O_4_-Cl	0.854	2.055	1.135	2.085	1.146	2.094
Co_3_O_4_-SO_4_	0.765	2.118	1.166	2.127	1.190	2.123
Co_3_O_4_-AC	0.850	2.061	1.111	2.085	1.129	2.097
Co_3_O_4_-NO_3_	0.817	2.070	1.120	2.084	1.138	2.092

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
