# Peer review of "Aggregation-Morphology-Dependent Electrochemical Performance of Co_3_O_4_ Anode Materials for Lithium-Ion Batteries"

_molecules, 2019, doi:10.3390/molecules24173149_

Round 1

Reviewer 1 Report

The manuscript describes the influence of aggregation morphologies on the electrochemical performance in nano-structured Co3O4 anode materials for Li-ion batteries. Although the authors have done various physico-chemical and electrochemical measurements and obtained data seems to be interesting, there are some places where the explanation seems insufficient. The Authors should be addressed following comments and questions before further processing:

Page 1: In introduction, it is better to describe the electrochemical reaction mechanism of Co3O4 for Li storage briefly, for the readers that not familiar to this material. Page 2, line 68 and 69: “rod like” may be not correct. “flower like” may be correct. Please check them. Page 3, line 121 to125: As the authors described, the shapes of precursors are strongly influenced by the kinds of Co salt. If possible, it is better to add the brief explanation of the mechanism to cause the morphology difference. Page 6, line 183: “oxide” should be revised into “oxidation”. Figure 7(b)-(d) and Figure 8(a)-(d): it is not clear that the “specific capacity” in the graphs is the gravimetric capacity for Li insertion process or Li extraction process. They should be clearly stated. Figure 7(b): In introduction, the authors stated that the theoretical capacity of Co3O4 is 890 mAh/g, but three samples (Co3O4-Cl, Co3O4-AC, Co3O4-NO3) show higher capacity than the theoretical capacity even after 100 cycles. In addition, the capacity of these three samples are increased gradually with the cycling. The author should explain in details about these results. Figure 10: What is the reason for the decrease in charge transfer resistance in the cell after the cycling? Figure 10: What is the origin for difference in charge transfer resistance in the cell cycled at 500 mA/g and 1000 mA/g?

Reviewer 2 Report

The authors describe the hydrothermal synthesis of various Co3O4 materials as high-capacity anodes for Li-Ion Batteries. Neither the material nor the approach are novel and the applicability of conversion-type materials in practise is quite low. The manuscript may be of scientific interest though and the approach large parts of the manuscript are well structured.

I think that the content of the manuscript does not match the scope of the MDPI Journal “Molecules” very well, despite the fact I found some similar publication in this journal. I would rather suggest a re-submission to the Journal “Batteries”.

In the introductory sentence, the authors emphasize the need for environmentally friendly rechargeable batteries, which the reviewer agrees with. However, I doubt that replacing the current state of the art anode material graphite with a costly and toxic Co-based anode is the most sustainable approach, despite the higher, theoretical gravimetric capacity of Co3O4 over graphite. Besides the issues linked to Cobalt, conversion-type materials are known to have poor Coulombic Efficiencies due to pronounced irreversible reactions (see Cabana, J.; Monconduit, L.; Larcher, D.; Palacín, M. R. Adv. Mater. 2010, 22 (35), E170.), especially in the 1st cycle, poor energy efficiency due to their high voltage hysteresis (see Meister, P.; Jia, H.; Li, J.; Kloepsch, R.; Winter, M.; Placke, T. Chem Mater 2016, 28 (20), 7203.) and pronounced volume changes upon operation. Further disadvantages are the very high average delithiation potential of Co3O4 >2V vs. Li/Li+, which lowers the energy density significantly (see Graphite ~0.1 V vs. Li/Li+; see [1] D. Andre, H. Hain, P. Lamp, F. Maglia, B. Stiaszny, J. Mater. Chem. A 2017, 5, 17174.) One crucial advantage of Co3O4 over graphite would be its high volumetric capacity (see Andre et al., Fig. 4a), which is not stated in your article).

- Please state the purity of your reagents, if possible.

- As far as I know, the EC:DEC ratio of electrolytes is measured by weight, not volume

- The red font in Fig 2a is barely readable. The contrast of the CoSO4 sample image is quite poor.

- Please state the areal mass loading (mg/cm2) and/or the areal capacity (mAh/cm2) of your electrodes

- The reviewer suggests to include full cell data vs. real cathode materials rather than half cell tests vs. metallic lithium to study a more realistic application scenario

- The galvanostatic cycling data in Figs 7+8 show very irregular behaviour. Please check cell reproducibility by cycling multiple cells and add error bar with standard deviation.

- Please state, why the experimental capacity values exceed the theoretical capacity of 890 mAh/g (weighing error?)

Round 2

Reviewer 1 Report

The authors have answered all questions from the reviewers and improved the manuscript by providing further data and discussion.

This manuscript can be accepted in its present form.

Author Response

There is nothing in the  manuscript that needs to be revised.

Reviewer 2 Report

Thank you for your revision and for addressing most of my request fast and satisfactory.

I only have two major concerns about the full cell study, which you have added (Fig. 12). First of all, the voltage window you chose is very unusual, as discharge voltages below 2.5-3.0V are rarely ever used in full cells. Thus, the data presented is not representative of a full cell and shows very odd voltage profiles. The upper part of the voltage profile resembles that of a Li/NMC full cell, however also in this case, it would not be discharged to those low potentials, which shows odd excess capacity during discharge. As a result, the NMC811 by far exceeds its practical capacity of ~200 mAh/g, which is also not realistic. Please comment.

Second of all, in full cell studies also the capacity balancing of the negative/positive electrode should be given. It is not very clear for the reader, whether the cathode is overbalanced or not, which would affect cycle life of the full cell.
